# Autism gene variants disrupt enteric neuron migration and cause gastrointestinal dysmotility

Kate E. McCluskey [1], Katherine M. Stovell [1], Karen Law [1],
Elina Kostyanovskaya [1], James D. Schmidt [1], Cameron R. T. Exner[1],
Jeanselle Dea [1], Elise Brimble [2], Matthew W. State [1], A. Jeremy Willsey [1] &
Helen Rankin Willsey [1,3] ✉

The co-occurrence of autism and gastrointestinal distress is well-established, yet the molecular underpinnings remain unknown. The identification of high-confidence, large-effect autism genes offers the opportunity to identify convergent, underlying biology by studying these genes in the context of the gastrointestinal system. Here we show that the expression of these genes is enriched in human prenatal gut neurons and their migratory progenitors, suggesting that the development and/or function of these neurons may be disrupted by autism-associated genetic variants, leading to gastrointestinal dysfunction. Here we document the prevalence of gastrointestinal issues in patients with large-effect variants in sixteen autism genes, highlighting dysmotility, consistent with potential enteric neuron dysfunction. Using *Xenopus tropicalis*, we individually target five of these genes (*SYNGAP1*, *CHD8*, *SCN2A*, *CHD2*, and *DYRK1A*) and observe disrupted enteric neuronal progenitor migration for each. Further analysis of *DYRK1A* reveals that perturbation causes gut dysmotility in vivo, which can be ameliorated by treatment with either of two serotonin signaling modulators, identified by in vivo drug screening. This work suggests that atypical development of enteric neurons contributes to the gastrointestinal distress commonly seen in individuals with autism and that serotonin signaling may be a productive therapeutic pathway.

Autism Spectrum Disorders (ASD) are neurodevelopmental disorders defined by atypical social interactions, repetitive behaviors, and restricted interests[1]. One of the most prevalent and impairing comorbidities of ASD is gastrointestinal (GI) distress, commonly presenting as gastric motility symptoms like constipation, diarrhea, or abdominal pain[2–4]. Despite this well-established comorbidity, the underlying molecular mechanisms remain unknown. The discovery of high-confidence (hc), large-effect autism genes[5] has opened the door to studying these genes in vivo in order to identify the molecular origins of this co-occurrence. In the central nervous system,

hcASD risk genes have been shown to converge in regulating the development of neural progenitor cells[6–13], motivating the hypothesis that these genes may also contribute to the development of the enteric nervous system (ENS), the neurons that control the GI tract. Indeed, the role of one of the first identified large-effect autism genes, *CHD8*, has been elaborated in zebrafish, where disruption of *chd8* caused reduced colonization of enteric neurons into the gut as well as gut dysmotility[14]. Consistently, it has been shown that autism genes are enriched in expression in human adult enteric neurons[15], the neurons controlling gastric motility[16]. Together, these findings

[1]Department of Psychiatry and Behavioral Sciences and the Weill Institute for Neurosciences, University of California San Francisco, San Francisco, CA, USA. [2]Citizen Health, San Francisco, CA, USA. [3]Chan Zuckerberg Biohub – San Francisco, San Francisco, CA, USA. ✉e-mail: helen.willsey@ucsf.edu

raise the possibility that the observed comorbidity may be due to atypical development of the ENS.

Here we document the prevalence of GI dysmotility in patients with pathogenic variants in hcASD genes, highlighting dysmotility, consistent with potential ENS dysfunction. These genes represent a wide swath of functional annotations and are highly pleiotropic, complicating efforts to establish direct links between molecular mechanisms and observed phenotypes[17–19]. A convergence neuroscience approach, where many genes are studied empirically in parallel and phenotypes and functions in common to multiple genes are considered more likely to be relevant to core biology, is one avenue to combat this challenge[17–19]. With this strategy in mind, we chose the model organism *Xenopus tropicalis* to study five hcASD genes representing disparate functional annotations in vivo to identify convergent phenotypes in the development of the ENS. These five genes include several with functional annotations in neurotransmission (*SYNGAP1*, *SCN2A*), gene expression regulation (*CHD8*, *CHD2*) and kinase activity (*DYRK1A*)[7]. These frogs have a simple diploid genome very similar to that of humans[20,21], an elaborate, coiled GI system[22], and the ability to perform unilateral genetic perturbations within an animal[6,20,23–25].

For all five gene perturbations individually (*SYNGAP1*, *CHD8*, *SCN2A*, *CHD2*, or *DYRK1A*), we observe defects in embryonic enteric neuron progenitor migration. We further focus on the hcASD gene *DYRK1A*, which encodes a kinase, and show that it is also required for gut motility in vivo. With a targeted drug screen we identify the selective serotonin reuptake inhibitor escitalopram and a serotonin receptor 6 agonist that can each individually ameliorate this dysmotility. Together, these results suggest that ENS developmental defects may be a convergent, contributing factor to the comorbidity between autism and GI distress and that increasing serotonin signaling may be a productive therapeutic strategy.

## Results

### hcASD gene expression is enriched in enteric neurons and their progenitors

Previously, hcASD gene expression has been shown to be enriched in adult human enteric neurons[15]. To assess hcASD gene expression in this tissue during development, we analyzed scRNA sequencing data from a range of stages during human prenatal gut development[26]. All 252 hcASD genes with an FDR < 0.1[5] were expressed in the data set. We used Module Score Analysis (AddModuleScore, Seurat R Package) to calculate the average expression of these genes and their relative expression enrichment across cell types, compared to a comparably expressed random control geneset (Supplementary Figs. 1a–b). hcASD gene expression enrichment was significantly higher in enteric neurons and their progenitors, enteric neural crest-derived cells (ENCCs), compared to all other cell types in the developing human prenatal gut (Fig. 1a, b, Supplementary Fig. 2, padj < 0.0001 and padj < 0.0001, Kruskal-Wallis test followed by Wilcoxon Rank-Sum tests with Bonferroni adjustment for multiple comparisons). These results suggest that hcASD gene expression is enriched in both mature and developing enteric neurons.

### Prevalence of GI dysmotility among patients with hcASD gene variants

Recently Simons Searchlight has compiled caregiver survey data for patients with large-effect variants in autism-associated genes and loci, as well as often for unaffected family members[27]. Within these data, we identified any patient with a genetic variant in one of the 252 hcASD risk genes. Nineteen genes were represented, with sixteen genes having more than ten affected individuals. Of these sixteen genes, seven are within the ten highest-associated hcASD genes (FDR < 0.01; *SYNGAP1*, *CHD8*, *ADNP*, *SCN2A*, *FOXP1*, *CHD2*, *GRIN2B*)[5]. All sixteen patient cohorts had caregiver-reported GI symptoms (average prevalence = 51.7%), while few unaffected family members

did (average prevalence = 0.4%) (Fig. 1c). To dive more deeply into the specific nature of these GI symptoms, we identified any patient in the Citizen Health medical record database with a genetic variant in a hcASD gene. Medical record data were available for 5 hcASD genes: *SYNGAP1* (n = 186 patients), *SCN2A* (n = 58 patients), *CHD2* (n = 41 patients), *SLC6A1* (n = 49 patients), and *STXBP1* (n = 80 patients). For all of these cohorts, over 80% of individuals had medical record diagnoses related to GI symptoms (Supplementary Dataset 1), consistent with caregiver reports for similar cohorts in Simons Searchlight described above. Of these diagnoses, GI dysmotility symptoms, particularly constipation, were reported in medical records for over 60% of individuals among all cohorts (Fig. 1d, e). Since GI motility is controlled by the ENS[28], these observations are consistent with the possibility that variants in these genes disrupt ENS development and/ or function, leading to gut dysmotility.

### hcASD gene variants disrupt ENS migration in vivo

Next we next aimed to examine the role of hcASD genes in the development of the ENS in vivo. ENS progenitors (ENCCs) are exclusively derived from the neural crest and migrate into the gut during embryonic development[29]. To study the role of hcASD genes in the development of these cells in vivo, we adapted the vertebrate diploid frog model *Xenopus tropicalis*, where parallel in vivo analysis of hcASD genes during embryogenesis is possible. Specifically, we visualized the ENCCs by whole-mount RNA in situ hybridization for their marker gene *phox2b*[16,30] (Fig. 2a) during embryogenesis following perturbation of hcASD genes by unilateral CRISPR/Cas9 mutagenesis, comparing the mutated half of the animal to the contralateral control side (Fig. 2b). We chose to perturb 5 of the 20 highest-associated hcASD genes[5] with disparate functional annotations (neurotransmission: *SYNGAP1* and *SCN2A*, chromatin regulation: *CHD2* and *CHD8*, and kinase activity: *DYRK1A*) by individually injecting Cas9 protein with an sgRNA into one cell of two-cell stage embryos. We have previously validated these sgRNAs in *X. tropicalis* for their loss-of-function efficiency[6,25]. We hypothesized that phenotypes observed in all of these five mutants are more likely to be relevant to the core underlying biology of the ASD/GI comorbidity, and less likely to be a byproduct of pleiotropy. The phenotype we observed for all five gene perturbations was a significant reduction in ENCC migration area, which was not observed in control animals injected with Cas9 protein and a sgRNA targeting a pigmentation gene *slc45a2* (Fig. 2c, d and Supplementary Fig. 3). This result was comparable for all five hcASD gene perturbations, irrespective of the gene's annotated cellular function, suggesting functional convergence in the process of ENCC migration.

### Depletion of *dyrk1a* disrupts neural crest development and ENCC migration

Next we elaborated this finding for *dyrk1a* in *Xenopus*, given our previous experience perturbing this gene within the central nervous system and the abundance of associated molecular tools[6,25,31,32]. We observed that *dyrk1a* is expressed in ENCCs during migration (Supplementary Fig. 4a) and persists in mature enteric neurons (Supplementary Figs. 4b, c, Supplementary Fig. 2), particularly at the primary cilium, an organelle required for ENCC migration[33]. As additional validation that our CRISPR-mediated disruption of *dyrk1a* was specific to this gene, we also perturbed *dyrk1a* with a validated translation-blocking morpholino[25,31] and observed the same phenotype as the CRISPR perturbation (Fig. 3a, b). This is particularly important since CRISPR/Cas9 gene editing in *Xenopus* F0 embryos often results in a mosaic, less severe phenotype[34], and it is possible that this mosaicism may result in unedited cells mitigating the migration phenotype since enteric neural crest-derived cells have been shown to migrate in chains[35].

ENCCs develop exclusively from the neural crest[29,36], and our group and others have observed *dyrk1a* expression in the *Xenopus* neural crest[25,31,32]. Therefore, this migration phenotype could be due to

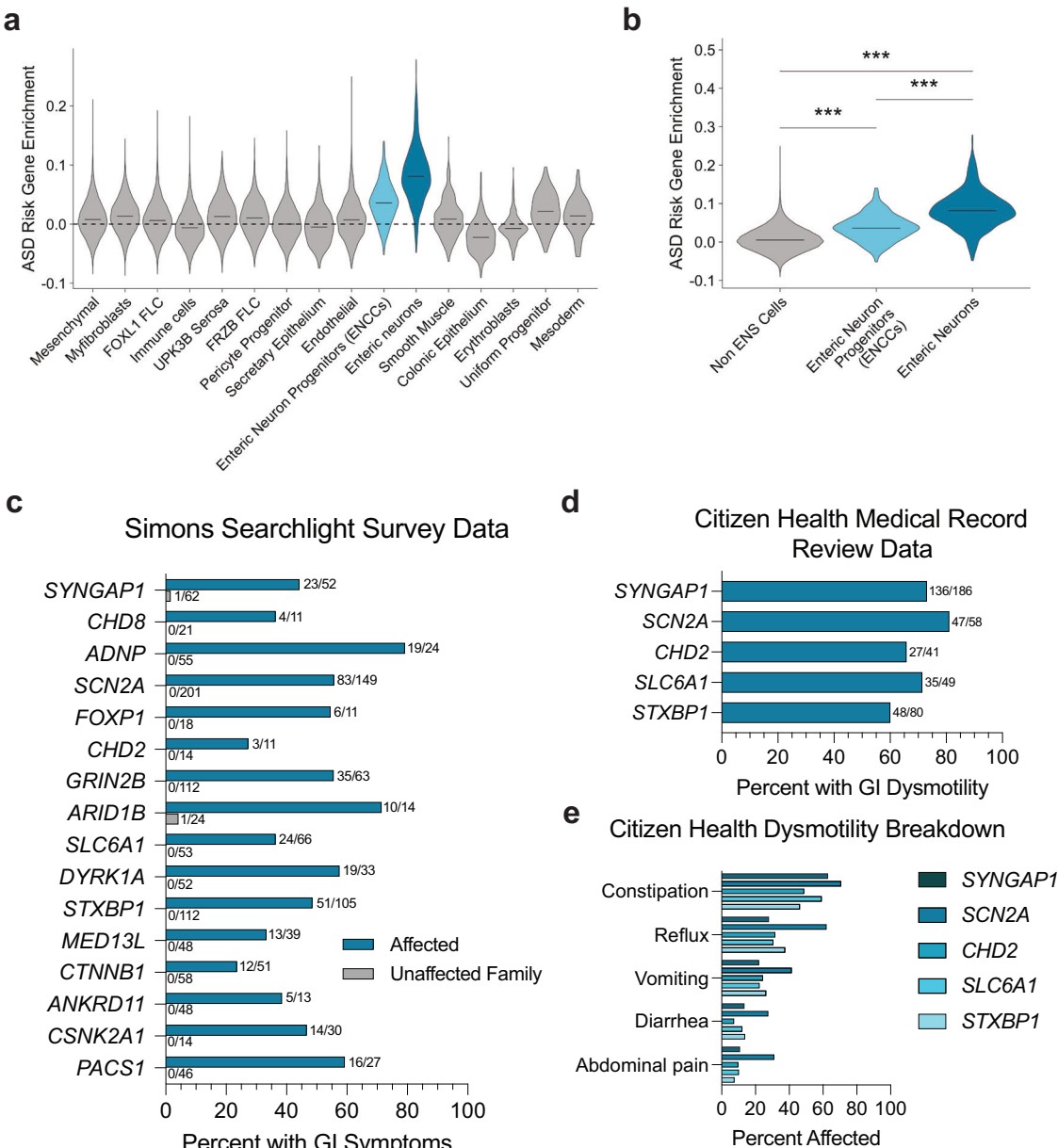

**Fig. 1 | hcASD gene expression is enriched in ENS cells and individuals with pathogenic variants in these genes experience GI issues. a** Enrichment of 252 hcASD genes is higher in Enteric Neuron Progenitors (ENCCs) and Enteric Neurons compared to all other cell types in single-cell RNA-sequencing data from the human prenatal gut[26]. **b** 252 hcASD genes are enriched in ENCCs and Enteric Neurons compared to all other Non-ENS cells in the human prenatal gut. A one-way Kruskal-Wallis test was performed, followed by Wilcoxon rank-sum tests and Bonferroni adjustment for multiple comparisons. Non-ENS vs ENCCs padj = 5.20e$^{-91}$, Non-ENS vs Enteric neurons padj = 0, ENCCs vs Enteric Neurons padj = 5.90e$^{-81}$. **c–e** The number of individuals affected and the total number of people surveyed is tallied at the end of each bar. **c** Simons Searchlight data documenting the percentage of affected individuals (teal bars) and their unaffected family members (gray bars) who reported GI issues in caregiver surveys. **d** Citizen Health medical record data showing the percentage of individuals with a *SYNGAP1*, *SCN2A*, *CHD2*, *STXBP1* or *SLC6A1* genetic variant with medical record diagnoses related to GI dysmotility. **e** Citizen Health medical record data by variant for dysmotility phenotypes including constipation, abdominal pain, and diarrhea.

earlier neural crest developmental defects. Indeed, by CRISPR or morpholino perturbation, we observed defects in the early neural crest marker *sox10*[37,38,] at NF stage 15 by whole-mount RNA in situ hybridization (Supplementary Fig. 5A). This indicates that Dyrk1a is required earlier in the embryo for neural crest development, a finding that is corroborated by a recent study[32]. To test whether other hcASD genes may also affect neural crest development, we knocked down the expression of *syngap1*, *chd8*, *scn2a*, or *chd2* and assayed *sox10* expression. We observed significant defects following depletion of *chd8* and *chd2*, a moderate effect for *syngap1*, and no effect for *scn2a* (Supplementary Fig. 6).

Next we aimed to separate the early requirement of these genes in neural crest development from any later roles in ENCC migration. To do this, we took advantage of the ability to conditionally inhibit Dyrk1a pharmacologically after neural crest specification is complete, at the onset of vagal neural crest migration[39] (Figs. 3a, b, Supplementary Fig. 5b). When we inhibited Dyrk1a with either TG003 or harmine[40] at the onset of migration (NF stage 25), ENCC migration was still disrupted and to a similar extent as it was by CRISPR perturbation, or compared to our positive control Ret inhibitor, a model of Hirschsprung's Disease[41,42] (Supplementary Figs. 5c–d). This was in contrast to the negative control treatments DMSO or moclobemide, a MAO

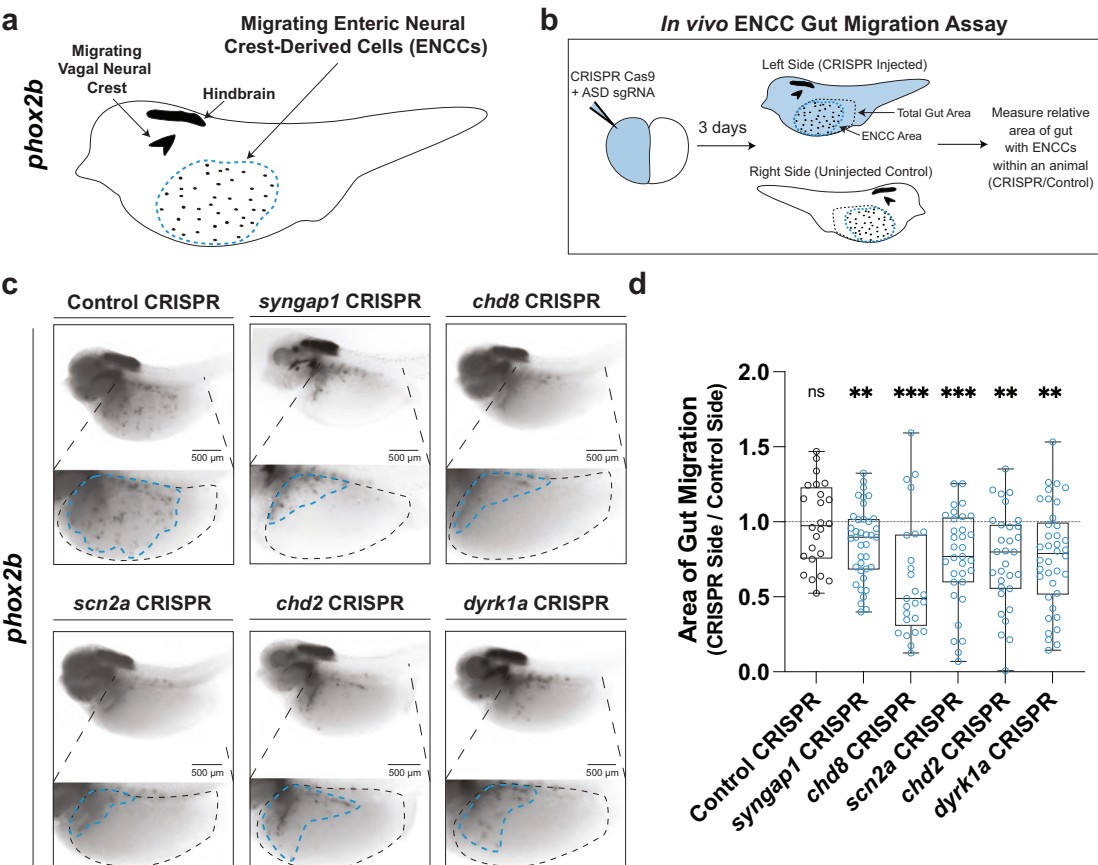

**Fig. 2 | hcASD gene depletion in vivo causes ENCC migration defects.**
**a** Schematic of *phox2b* staining in NF stage 40 animals to mark enteric neural crest-derived cells (ENCCs, enteric neuron progenitors) in the gut, circled by a blue dashed line. *phox2b* also labels the hindbrain region and other migrating vagal neural crest cells. **b** Unilateral mutants were made by injecting the Cas9 protein and an sgRNA targeting an hcASD gene into one cell at the two-cell embryonic stage. Three days later, injected embryos were stained and ENCC area and gut area was measured on each side of the animal and compared to quantify relative gut area (CRISPR side / Control side) of migration. **c** Individual CRISPR mutagenesis of hcASD genes *syngap1, chd8, scn2a, chd2,* or *dyrk1a* reduce the area of ENCC migration in the gut compared to control mutagenesis of pigmentation gene

*slc45a2*. **d** Area of gut migration quantification by target gene. Control in gray, hcASD genes in blue. A two-way Kruskal-Wallis test was performed followed by Wilcoxon matched-pairs signed rank test to compare the CRISPR side to the control side within each animal and a Holm-Šídák test to adjust for multiple comparisons. Control ($N$ = 24, padj = 0.9248), *syngap1* ($N$ = 38, padj = 0.0046), *chd8* ($N$ = 25, padj = 0.001), *scn2a* ($N$ = 33, padj = 0.001), *chd2* ($N$ = 31, padj = 0.0011), *dyrk1a* ($N$ = 38, padj = 0.0011). All samples are independent biological replicates from the same mating pair. In the box plot, whiskers show minimum and maximum values, box represents the 25th and 75th quartiles, and the center line describes the median. Source data are provided as a Source Data file.

inhibitor that controls for a potential off-target effect of harmine[43]. Together, these results suggest that *dyrk1a*, while required for early neural crest development, is also independently required for ENCC migration in *Xenopus*.

**Depletion of dyrk1a decreases gut motility in vivo**
We next assessed whether inhibition of Dyrk1a during development affects gut motility in vivo. As seen above, gut dysmotility in the form of constipation is common among patient cohorts for several hcASD risk genes. To assess gut transit in *Xenopus*, we developed an in vivo gut motility assay where mature tadpoles (NF stage 47) were fed food with 6 μm fluorescent beads for 2 h, rinsed out of the food and placed in baskets in 6-well plates to defecate for 3 h, and then taken out by removing the baskets while leaving the excrement and fluorescent beads behind (Fig. 3c). The plates were then imaged with a fluorescence microscope and the number of fluorescent beads per well was counted in Fiji. Each well contained 20 tadpoles and each condition was completed in triplicate. To test the role of Dyrk1a in gut motility in vivo, we raised animals in the Dyrk1a inhibitor TG003 or in control DMSO solution (starting at NF stage 20) and then performed the in vivo gut motility assay at mature tadpole stage (NF stage 47).

There was a significant decrease in the number of fluorescent beads excreted in animals treated with the Dyrk1a inhibitor, compared to the DMSO control-treated tadpoles ($p$ = 0.0091, one-tailed t test with Welch's correction) (Fig. 3d, e). This result suggests that developmental Dyrk1a inhibition in vivo causes gut dysmotility in *Xenopus* tadpoles.

**Serotonin reuptake inhibitor treatment rescues gut dysmotility following Dyrk1a inhibition**
Serotonin signaling is known to control enteric neuron activity and gut motility[44]. Therefore, we selected 8 FDA-approved drugs that impact serotonin signaling at various points in the pathway (Fig. 4a) and tested whether acute treatment alters gut movement with our in vivo gut motility assay in unmanipulated mature tadpoles. Of these drugs, at the concentration we tested (10 μM), only the selective serotonin reuptake inhibitor (SSRI) escitalopram oxalate was able to promote gut motility greater than 1 standard deviation compared to negative control treatment DMSO (Fig. 4b). Escitalopram oxalate (also known as Lexapro) is a common SSRI that interacts with SERT, the serotonin transporter, to inhibit the reuptake of serotonin into the presynaptic cell, thereby prolonging serotonin pathway activation (Fig. 4a). This

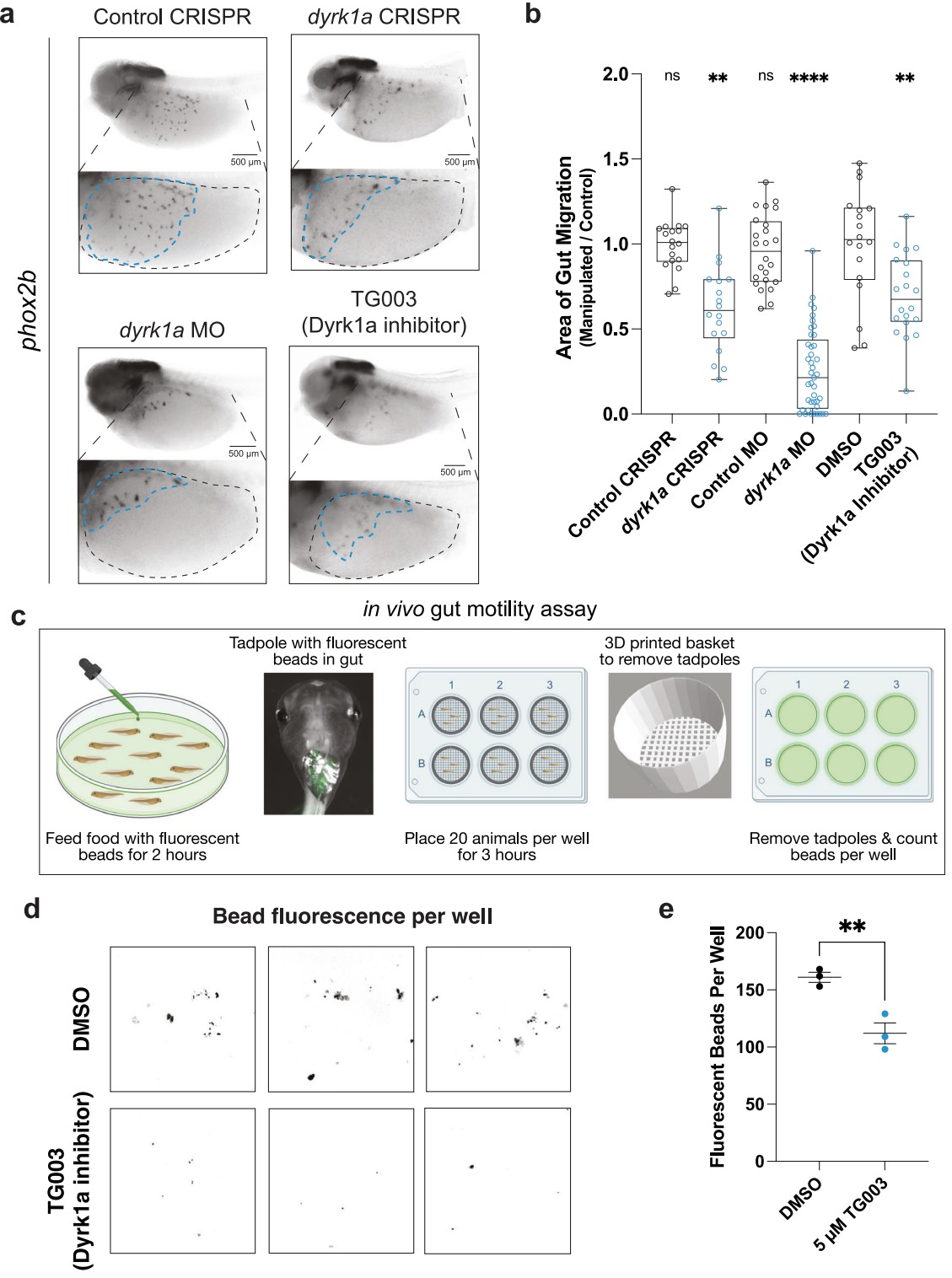

**c** *in vivo* gut motility assay

**d** **Bead fluorescence per well**

result is consistent with previous studies in which activating or prolonging active serotonin signaling promotes gut motility in mammals[45–48].

Next we tested whether this SSRI could acutely rescue the effects of developmental Dyrk1a inhibition on gut motility. In addition to this SSRI, we concurrently tested a more specific activator of serotonin signaling, an agonist for serotonin receptor 6 (5-HTR6) (WAY-181187[49]). We included this drug because both Dyrk1a and serotonin receptor 6 specifically localize to primary cilia[50] (Supplementary Figs. 4c–d). To test these compounds, we raised tadpoles in DMSO or TG003 (Dyrk1a

inhibitor) and performed our in vivo gut motility assay, adding 10 μM escitalopram oxalate or WAY-181187 to a portion of the TG003-treated animals acutely during the 3 h excretion portion of the assay. Acute exposure to either of these compounds during the excretion portion of the assay (3 h) rescued the decreased motility phenotype of the Dyrk1a inhibitor-treated animals, restoring excretion to that comparable to the DMSO control (Fig. 4c, d). Therefore, these results suggest that Dyrk1a-associated gut dysfunction can be ameliorated by acute treatment with an SSRI or, more specifically, with a 5-HTR6 agonist. Together, our work suggests a model (Fig. 4e) in which ASD genetic

**Fig. 3 | hcASD gene *dyrk1a* perturbation reduces ENCC migration and decreases gut motility in vivo. a** *dyrk1a* perturbation through CRISPR injection, morpholino (MO) injection, or Dyrk1a inhibitor (TG003) treatment all reduce the area of ENCC migration in the gut compared to control CRISPR, control MO, or vehicle control (DMSO). **b** Area of gut migration quantification by condition. For injection comparisons, a two-tailed Wilcoxon matched-pairs signed rank test to compare the CRISPR or morpholino side to the control side within each animal was performed. Control CRISPR (N = 18, p = 0.7987), *dyrk1a* CRISPR (N = 18, p = 0.0032), Control MO (N = 24, p = 0.1208), *dyrk1a* MO (N = 41, p < 0.0001). For small molecule treatment, a two-tailed Wilcoxon rank sum test was performed to compare DMSO (N = 18) to 5 μM TG003-treated (N = 20) embryos, p = 0.0026. All samples are independent biological replicates from the same mating pair. In the box plot whiskers show minimum and maximum values, box represents the 25th and 75th quartiles, and the center line

describes the median. **c** Schematic of the in vivo gut motility assay. At NF stage 47, tadpoles were fed food with fluorescent beads for 2 h, then placed in a 6-well plate in 3D-printed baskets with 20 animals/well for 3 h. Tadpoles are then removed by taking out baskets and leaving excrement with fluorescent beads behind, after which the plates were imaged to count the number of beads per well. Created in BioRender. McCluskey, K. (2025) https://BioRender.com/w82f112. **d** Developmental inhibition of Dyrk1a (TG003-treated beginning at NF 20) results in decreased fluorescent bead excretion compared to vehicle control (DMSO). Representative images of fluorescent beads (false-colored black). **e** Number of fluorescent beads per well counted for DMSO (N = 3 wells, 20 tadpoles each) and TG003 (N = 3 wells, 20 tadpoles each) (Dyrk1a inhibitor) wells. A one-tailed *t* test with Welch's correction was used to compare DMSO to 5 μM TG003, p = 0.0091. Data are presented as mean values +/− SEM. Source data are provided as a Source Data file.

variants cause atypical ENCC development, leading to GI dysmotility, and suggests serotonin signaling as a potential therapeutic pathway.

## Discussion

The co-occurrence of autism and GI issues is well-established[2], yet the molecular mechanisms underlying this comorbidity remain unclear. Here we document the prevalence of GI issues in individuals with variants in hcASD genes, particularly highlighting constipation, a symptom of GI dysmotility. Our group and others have shown that many hcASD genes converge to regulate neuronal development in the central nervous system[6–13,51], motivating us to consider whether these genes also contribute to the development of the ENS. Consistently, we observe that hcASD gene expression is enriched in both enteric neurons and their migrating progenitor cells during human embryonic development. Therefore, to test this hypothesis, we use a vertebrate in vivo model to individually perturb five hcASD genes with disparate functional annotations (neurotransmission, chromatin regulation, kinase activity). We observe disrupted ENCC migration for each, suggesting that atypical ENCC migration may contribute to GI dysmotility in ASD. We further investigate the role of Dyrk1a in gut motility and observe that Dyrk1a perturbation leads to a decrease in excretion that can be restored by acute exposure to agonists of serotonergic signaling.

This work adds to a growing literature suggesting that variants in ASD genes perturb ENS development[3,52], consistent with work done in zebrafish for *CHD8*[14] and in mice for *FOXP1*[53] and *PTEN*[54]. While a fish study of *SHANK3* did not observe a significant change in gut neuron number[55], this group did identify disrupted serotonergic signaling in the gut, consistent with our findings that manipulating serotonin signaling could be a potential therapeutic avenue. Indeed, the role of serotonin signaling in gut motility is well-established[45–48], so this pathway could be productively exploited in the context of ASD more broadly for treating GI issues. Moving forward, other key questions remain including how ASD gene variants affect enteric neuron number, differentiation, organization, morphology, and activity in mature animals. In particular, the role of cell death in these phenotypes will need to be explored, as we have previously shown that DYRK1A inhibition in the central nervous system causes apoptosis[25]. Additionally, our work here characterizing loss of function for these genes could be complemented by the use of heterozygous germline mutant animals to recapitulate the haploinsufficiency present in individuals with pathogenic de novo variants in these genes. Finally, intersecting these findings regarding ENS development with other proposed mechanisms of GI dysmotility in ASD, particularly since ENS developmental defects are known to impact the composition of the gut microbiome[56,57].

Impaired ENCC migration is a central observation of our study, consistent with previous work in an assembloid model suggesting that neuron migration may be a convergent process disrupted in neurodevelopmental disorders[58]. Interestingly, we observed that ENCC migration is affected by perturbation of genes known for their roles in neurotransmission (*SYNGAP1*, *SCN2A*). Similar to our previous findings

in *Xenopus* central nervous system development[6] that have been recently validated in human cortical organoids[13], this work suggests that these genes may have additional roles during ENS development beyond neurotransmission. Recently, several of these neurotransmission proteins have been shown to localize to the cilium[59], an organelle known to be required specifically for ENCC migration into the gut[33]. Here we show that Dyrk1a localizes to the primary cilium in enteric neurons in *Xenopus* (Supplementary Fig. 4c). It is also known that serotonin receptor 6 (5-HTR6) specifically localizes to the cilium in neurons[50]. Here we show that acute exposure to a 5-HTR6 agonist is able to increase gut motility and rescue a Dyrk1a model of dysmotility. Given the role of serotonin in gut motility and development[60], a productive area of future investigation will be to understand how other ASD gene variants perturb primary cilia in the development and function of the ENS and whether SSRIs can affect gut motility in those genetic contexts as well.

In summary, this work documents the prevalence of gut dysmotility, particularly constipation, in many individuals with large-effect variants in hcASD genes. Disruption of five hcASD genes convergently impacts ENCC migration in *Xenopus*, and, based on the study of Dyrk1a, this disruption may underlie the gut dysmotility seen in affected individuals. It will be important future work to determine the extent to which perturbations of other hcASD genes also result in gut dysmotility, and whether serotonin activation can also ameliorate any observed phenotypes. If this finding holds for many hcASD genes, investigating the therapeutic potential of serotonin activation for restoring gut motility in affected individuals may be a productive endeavor.

## Methods
### Simons searchlight and citizen health patient data analysis
Research with Simons Foundation Autism Research Initiative (SFARI) Simons Collection (Searchlight) data and with Citizen Health medical record data was determined to be non-human subjects research by UCSF IRB office, IRB#23-39-079. Unaffected family and affected individual GI data was accessed through the Simons Searchlight Single Gene Dataset v8.0. All data was used from families with any genetic variant in a hcASD gene, as defined by Fu et al. 2022[5] with an FDR < 0.1. Patient GI data was additionally accessed from Citizen Health. Patients were included if they possessed a genetic variant that was classified on their genetic report as 'pathogenic', 'likely pathogenic', or 'variant of uncertain significance'. Individuals with 'benign' variants in a hcASD gene were excluded. Individuals with a 'variant of uncertain significance' in a hcASD gene in addition to a 'pathogenic' variant in a non-ASD gene were also excluded. Data analysis and visualization were performed in Prism (v.10.1.1) software.

### Human prenatal intestine scRNA-sequencing analysis and enrichment scoring
First, scRNA-seq FASTQ files of human prenatal intestine (ArrayExpress: E-MTAB-9489[26]) were downloaded from ArrayExpress and

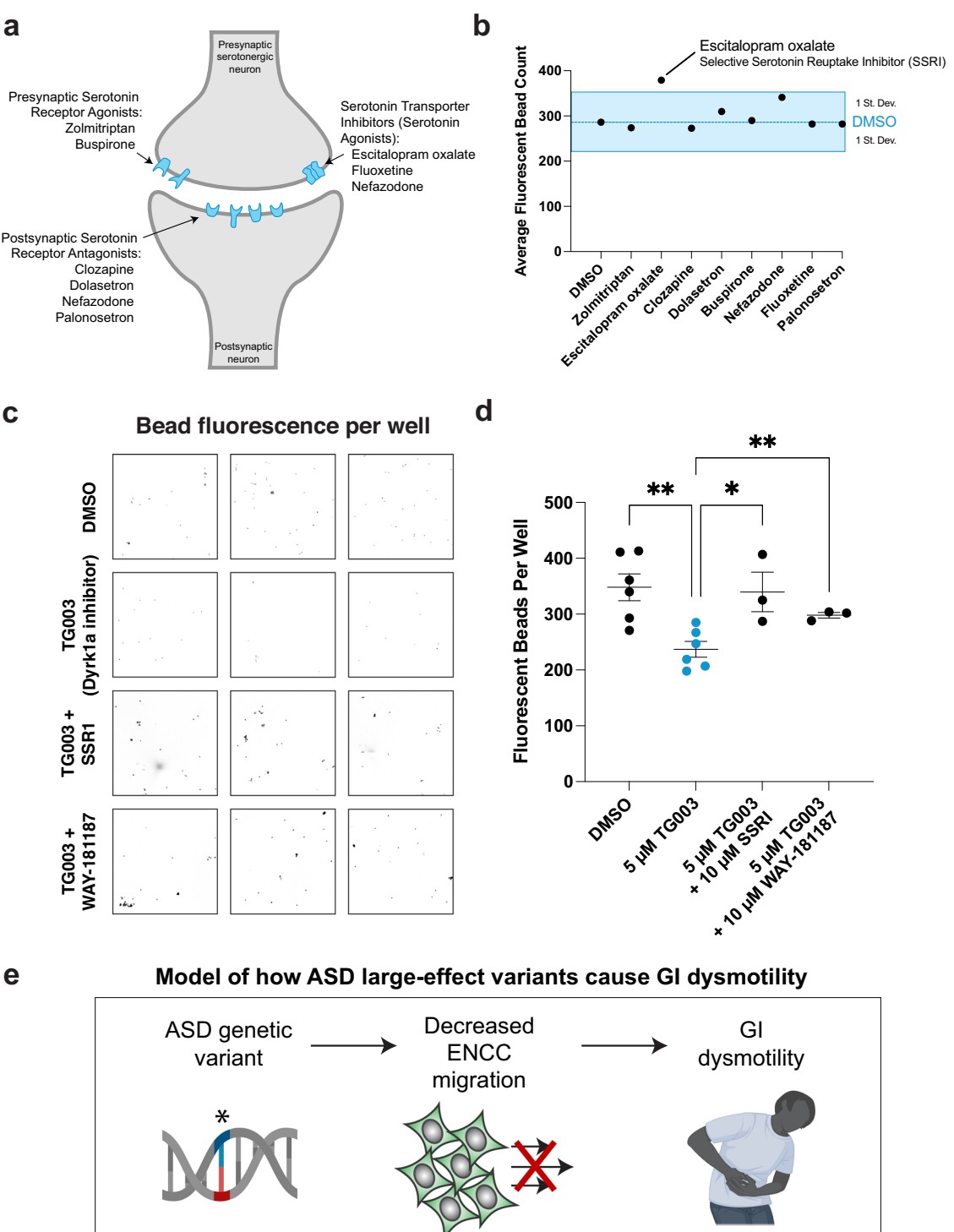

**Fig. 4 | Dyrk1a-associated gut dysmotility is ameliorated by acute exposure to an SSRI or 5-HTR6 agonist. a** Schematic of a serotonergic synapse labeling the presumed location of action for the drugs tested. **b** Acute exposure of wildtype tadpoles to escitalopram oxalate (selective serotonin reuptake inhibitor, SSRI) alone increases the average number of fluorescent beads excreted by more than one standard deviation compared to the average DMSO treatment, blue. **c** Acute treatment of an SSRI (10 μM escitalopram oxalate) rescued, while a 5HTR6 agonist (10 μM WAY-181187) partially rescued the decreased fluorescent bead excretion from developmentally inhibited Dyrk1a (5 μM TG003 starting at stage NF 20) animals. **d** Fluorescent beads per well quantified for each treatment condition. A one-way ANOVA was performed followed by unpaired one-tailed *t* tests with welch's

correction. Compared to TG003 treatment ($N$ = 6 wells, 20 tadpoles each), DMSO ($N$ = 6 wells, 20 tadpoles each $p$ = 0.002), TG003 + SSRI ($N$ = 3 wells, 20 tadpoles each $p$ = 0.0423), TG003 + WAY-181187 (N = 3 wells, 20 tadpoles each $p$ = 0.0033). Compared to DMSO, TG003 + SSRI $p$ = 0.4264 and TG003 + WAY-181187 $p$ = 0.0462. All samples are independent biological replicates from the same mating pair. Data are presented as mean values +/– SEM. Source data are provided as a Source Data file. **e** Model of how hcASD gene large-effect variants could contribute to GI dysmotility. An hcASD genetic variant leads to perturbed ENCC migration and ultimately GI dysmotility. Created in BioRender. McCluskey, K. (2025) https://BioRender.com/n91b055.

processed by Cell Ranger (v6.0.1, GRCh38). Following the standard Seurat (v5.0.3) analysis pipeline in R (v4.4.0), we read the gene-barcode matrix outputs of all samples and filtered out low quality cells that had >10% mitochondrial counts and unique feature counts over 2500 or less than 200, leaving 39,095 cells. Normalization was then done with the gene-barcode matrix with method LogNormalize that normalized each cell by total expression and multiplied by the default scale factor (10,000).

2000 highly variable genes of each sample were identified with function FindVariableFeatures then all samples were integrated with method IntegrateData to form a combined Seurat object for downstream analysis. Integrated object was then scaled and Principal Component Analysis was performed for dimensional reduction. FindNeighbors and FindClusters were used to compute cell clusters of the object. Next, identification of cell clusters was performed using gene markers from Elmentaite et al. 2020[61] in addition to ENCC markers (*SOX10*, *FOXD3*, *PHOX2B*) and enteric neuron markers (*TUBB3*, *ELAVL4*, *RET*) to finalize cell annotations.

To assess enrichment of hcASD genes, AddModuleScore function was performed for the 252 hcASD genes from Fu et al. 2022 and compared against a control geneset with equivalent average expression across all cells. Each cell received a module score and was then grouped based on cell annotation as described above. A cell expressing hcASD genes higher than the control geneset received a score > 0, while a cell expressing the control geneset higher than the hcASD genes had a score of <0. Cells were then grouped into Non-ENS cells, ENCCs, and enteric neurons based on the above clustering. Next we tested whether ENCCs and enteric neurons had significantly higher hcASD gene expression enrichment scores, compared to all other cells by a Kruskal-Wallis test, followed by Wilcoxon rank-sum tests and Bonferroni adjustment for multiple comparisons. The enrichment of these genes in each cluster or group was plotted using VlnPlot.

### *Xenopus* husbandry

*Xenopus laevis* or *tropicalis* adults originated from the National Xenopus Resource Center[62] and the Khokha Lab. Animals used were of wildtype lines of both sexes. Animals were housed in a recirculating system and were maintained in accordance with approved UCSF IACUC protocol AN199587-00A. Ovulation was induced with human chorionic gonadotropin (Sigma CG10), and embryos were collected and manipulated according to standard procedure[63]. Gut motility experiments were done in *Xenopus laevis*, while migration experiments were done in *Xenopus tropicalis*. The *Xenopus* community resource Xenbase was referenced daily[64].

### *Xenopus tropicalis* microinjections and loss of function

A Narishige micromanipulator, Parker or Narishige Picospritzers, and Zeiss Stemi508 microscopes were used to microinject reagents into 1 cell of 2-cell stage embryos. For morpholino injections, 2 nl injection volume of each condition was measured using an eyepiece micrometer, delivering 2 ng of the morpholino per blastomere. Embryos were injected and grown at 20–25 °C in 1/9 Modified Ringer's (MR) solution. Published and validated translation-blocking *dyrk1a* morpholino (5′ TGCATCGTCCTCTTTCAAGTCTCAT 3′)[25,31] was compared against a standard control morpholino (5′ CCTCTTACCTCAGTTACAATTTATA 3′, GeneTools). For the supplemental *sox10* experiment (Fig. S5), translation-blocking morpholinos for *syngap1* (5.5 ng, 5′ GAGCATAGAACATCATTCCACAGCT 3′, GeneTools), *chd8* (3.2 ng, 5′ CCAGCCTGTGAGAGAAGATAGTAAT 3′, GeneTools), *scn2a* (3.2 ng, 5′ ATTCTGGAGGATTACTCAAGGTCAT 3′, GeneTools), and *chd2* (3.2 ng, 5′ GGTTTATCCTCATTCCTCATCATTG 3′)[65] were compared against 5.5 ng of the standard control morpholino.

For CRISPR, we synthesized (EnGen, NEB E3322S) and purified (Zymo R1018) validated sgRNAs[6] and injected 400 pg of sgRNA and 2.24 ng of Cas9-NLS protein (UC Berkeley MacroLabs[66]) into 1 cell of

2-cell stage *X. tropicalis* embryos. The exceptions to this are *syngap1* and corresponding *slc45a2* control where embryos were injected with 800 pg of sgRNA and 4.48 ng of Cas9-NLS protein, since the *syngap1* sgRNA is of lower mutational efficiency. Injection volume was measured using an eyepiece micrometer and embryos were incubated at 27 °C for 1 h immediately after injection and then grown at 20–25 °C in 1/9 MR solution.

### *Xenopus* whole-mount RNA in situ hybridization

All embryos were fixed in MEMFA for 1 h and stained according to Willsey et al. 2021[67] using anti-DIG (1:3000, Sigma 11093274910) and BM Purple (Sigma 11442074001). *phox2b* probe plasmid was a kind gift from the Harland lab (IMAGE clone #8956276, probe synthesis with T7 enzyme and EcoR1 restriction enzyme), and *sox10* probe was previously described[6]. The stainings were performed in a high-throughput basket format with 200 μm mesh in 3D-printed racks[67,68]. All embryos were imaged on a Zeiss AxioZoom.V16 with a 1x objective and extended depth of focus processing in Zeiss Zen software.

For hybridization chain reaction (HCR)[69,70], embryos were stained according to Willsey et al. 2021[67], with an additional step of heating probes to 95 °C for 90 seconds and cooling to room temperature before use. *dyrk1a* and *phox2b* probes were custom ordered through Molecular Instruments and designed to target the *X. tropicalis* genes. Embryos were imaged on a Zeiss 980 LSM with a 63x oil objective, Airyscan processed and Maximum Intensity Projection performed in Zen or on a Zeiss AxioZoom.V16 with a 1x objective and extended depth of focus for full-embryo imaging.

### Drug treatment

Dyrk1a inhibitor (TG003, Sigma T5575) and MAO inhibitor control (Moclobemide, Sigma M3071) were resuspended in DMSO at 10 mM. Dyrk1a inhibitor (harmine, Sigma 286044) and Ret inhibitor (PP1, Sigma P0040) were resuspended in DMSO at 1 mM. Embryos were treated with drug or an equal volume of DMSO, diluted in 1/9 MR, at NF stage 25 and fixed at NF stage 40 for RNA in situ hybridization staining. Media was not refreshed while embryos were growing. For the serotonin gut motility screen (see below), selected drugs from an FDA-approved drug library (Enzo Life Sciences BML-2843-0100) were tested at 10 μM alongside an equal volume of DMSO diluted in 1/3 MR.

### Gut motility assay

All gut motility assays were performed in *Xenopus laevis*, since pilot studies showed they excrete more than *X. tropicalis* (they are larger animals), which was enough to be able to reliably quantify bead number per well from 20 animals. Since these experiments are all drug treatments and not genetic perturbations, the advantages of the *X. tropicalis* diploid genome are less critical. *Xenopus laevis* embryos were collected and raised for 9 days at room temperature until NF stage 47. At the beginning of the gut motility assay, tadpoles were moved into 15 cm dishes containing 20 mL of 1/3 MR. Food was prepared by resuspending 1 g of sera micron (Sera 00720) in 45 mL of 1/3 MR with 30 μL of 6 μm fluorescent bead solution (Polysciences 18862-1) added, similarly to what has previously been used to assay gut motility in zebrafish[55]. 2.5 mL of the food suspension was added to each 15 cm dish for each condition. Tadpoles were left to feed for 2 h and then rinsed in a homemade mesh (Spectra Mesh 100 μm and 200 μm, 146488 and 146487) trough in a glass dish (Grainger 900203). Using a vacuum, media was removed while fresh 1/3 MR was added until the media was clear of food and beads. Tadpoles were moved from the mesh trough using a plastic pipette into a fresh 15 cm dish. They were then transferred to a 6-well plate with homemade 3D-printed baskets with lattice bottoms inserted into each well. 20 tadpoles were allotted into each basket. After defecating for 3 h, baskets with tadpoles were removed from each well, leaving excretion with fluorescent beads in the well. All

plates were left in the dark at 4 °C overnight to let all matter sink to the bottom of the plate. Then, plates were tile-imaged in the same position on a Zeiss AxioZoom.V16 with a 1x objective and analyzed in Fiji.

When raising animals in the Dyrk1a inhibitor, TG003 or equivalent volume of DMSO was added to 1/3 MR to make it a 5 μM TG003 solution. 75 NF stage 20 *X. laevis* embryos were treated for each condition. For the serotonin screen, tadpoles were raised only in 1/3 MR until the 3-h defecation period, at which point they were acutely exposed to 10 μM of each compound individually. For the rescues, animals were grown in DMSO or 5 uM TG003 starting at stage 20. At stage 47, animals were removed from drug solution and fed beads in 1/3 MR as above. Following feeding, animals were left to defecate either in 1/3 MR or in 10 μM SSRI or 5-HTR6 drug solution in 1/3MR.

### Image processing and statistical analyses
All images were processed in FIJI (NIH) and arranged in Illustrator (Adobe). For ENCC quantification, the area of ENCCs and total gut area were both measured by drawing with the free-hand selection tool and quantified with the measure function and experimenter was not blinded to experimental condition. The ENCC area was normalized to the total gut area. Tests for normality were performed in Prism (GraphPad) followed by an ANOVA or Kruskal-Wallis test as appropriate. For unilateral mutagenesis, Wilcoxon matched-pairs signed rank tests were performed to compare the uninjected and injected sides of each embryo per condition and then adjusted for multiple comparisons with the Holm-Šídák method when appropriate. For small molecule experiments, a two-tailed Wilcoxon rank sum test was performed to compare treatment conditions to control conditions, or a Kruskal-Wallis test with a Dunn's test to correct for multiple comparisons when appropriate.

For gut motility image analysis, tiled images were analyzed in FIJI with a macro to Find Maxima (Prominence > 60, Output type: Count) in each well of a 6-well plate. Counts were imported into Prism and standard deviation was calculated for control treatments. For comparing between DMSO, TG003 and rescue treatments, all conditions passed normality and a one-way ANOVA was performed followed by unpaired one-tailed *t* tests with Welch's correction.

### Cell culture
84,000 RPE-1-hTERT (ATCC CRL-4000) cells per well were seeded into 24-well Corning cell culture dishes in DMEM F12 (Thermo Fisher Catalog # 11320-082) supplemented with 10% Fetal Bovine Serum, incubated at 37 °C, in 5% $CO_2$ for 24 h. After 24 h, cell media was replaced with reduced serum media OptiMEM (Thermofisher 31985062) to induce serum starvation. 24 h later, cells were fixed with 4% PFA for 10 minutes, washed with three times PBS, and stored at 4 °C for further processing.

### Immunofluorescence staining
For NF stage 45 samples, Immunostaining was performed according to Willsey et al. 2021[67] with the omission of the bleaching step as it was found to add bubbles that affected GI morphology. Phalloidin (1:400, Life Tech A34055) and DAPI (1:400) were added during the secondary antibody step. Primary antibodies used were against Dyrk1a (1:100, R&D Systems AF5407) or Acetylated α-Tubulin (1:200, Abcam ab179484). Secondary antibodies (Abcam ab150177, Thermofisher A32733) were diluted at 1:250. Embryos were imaged on a Zeiss 980 LSM with a 63x oil objective, Airyscan processing and Maximum Intensity Projection were performed in Zen.

RPE-1 cells were permeabilized with PBST (1X PBS at pH 7.4 containing 0.2% Triton X-100) three times for 5 minutes and blocked with 2% BSA (Sigma-Aldrich A70906 in PBST) for 5 h at room temperature. Primary antibodies were diluted into 2% BSA and cells were incubated at 4 °C overnight. Antibodies used were ARL13B (1:2000, ProteinTech

17711-1-AP) and Dyrk1a (1:250, R&D Systems AF5407). Following primary antibody incubation, cells were washed three times for 5 minutes in PBST, protected from light. Secondary antibodies (Thermo A32732 or Thermo A32728) were diluted at 1:1000 in 2% BSA, and DAPI was included in this secondary incubation at 1:1000. Cells were incubated in diluted secondary antibodies for 1 h at room temperature, protected from light. Stained cells were washed with PBST three times for 5 minutes and with 1X PBS two times; cells were stored at 4 °C. Cells were imaged on a Zeiss 980 LSM with a 63x oil objective. Maximum intensity projection was performed in FIJI.

### Reporting summary
Further information on research design is available in the Nature Portfolio Reporting Summary linked to this article.

## Data availability
The hcASD gene list is publicly available[5]. Publicly available human fetal gut scRNA-seq data is accessible from ArrayExpress: E-MTAB-9489. Citizen Health medical record data is available with proper IRB approval from Citizen Health (www.citizen.health). Approved researchers can obtain the Simons Searchlight population dataset described in this study by applying at https://base.sfari.org. Source data are provided with this paper.

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

## Acknowledgements

We are grateful to all of the families at the participating Simons Searchlight sites as well as the Simons Searchlight Consortium, formerly the Simons VIP Consortium. We appreciate obtaining access to Simons Searchlight phenotyping data on SFARI Base. We thank Nolan Wong, Louie Ramos, and UCSF LARC for animal care; Juan Arbelaez and Ethel Bader for lab maintenance and support; Hasan Alkhairo for coding discussions; Edivinia Pangilinan for expert technical help; and Ashley Clement, Gigi Lopez, Sonia Lopez and Linda Chow for administrative support. This work would not be possible without daily reference to the *Xenopus* community resource Xenbase (RRID:SCR_003280) and expertise and frog resources from the National *Xenopus* Resource (RRID:SCR_013731). This work was supported by a grant from the National Institutes of Mental Health (U01MH115747 to A.J.W. and M.W.S.) and an investigator award from the Chan Zuckerberg Biohub - San Francisco (to H.R.W.).

## Author contributions

Conceptualization: K.E.M., H.R.W., A.J.W., M.W.S. Methodology: K.E.M., H.R.W., A.J.W., K.M.S., K.L., C.R.T.E. Validation: K.E.M. Formal Analysis: K.E.M., K.L., E.B. Investigation: K.E.M., E.K., J.D.S., C.R.T.E., J.D. Resources: H.R.W., A.J.W., M.W.S., E.B. Writing – Original Draft: K.E.M. Writing – Review & Editing: H.R.W., K.E.M., A.J.W., M.W.S. Visualization: K.E.M., H.R.W., E.K. Supervision: H.R.W., M.W.S., A.J.W. Project Administration: H.R.W. Funding Acquisition: A.J.W., H.R.W., M.W.S.

## Competing interests

E.B. is a salaried employee of Citizen Health, with vested and unvested stock options. The remaining authors do not have any competing interests.
