## [Transparent Peer Review file · Nature Communications]

Autism gene variants disrupt enteric neuron migration and cause gastrointestinal dysmotility

Corresponding Author: Dr Helen Willsey

Version 0:

Reviewer comments:

Reviewer #1

(Remarks to the Author)

This paper by McCluskey and colleagues hits on all the aspects of a great paper. From human gene-phenotype data to mechanistic and behavioral data in an effective animal model. I believe this manuscript should be published in this journal with no hesitation since it is original and of great significance to understanding comorbidities associated with autism. Moreover, this work also contributes to our basic understanding of innervation of the digestive tract. With a few simple experiments and modifications to the writing I believe this paper to be of high impact and a great addition to the autism and developmental biology fields. I have made some suggestions that could elevate, add a bit more depth and help readers interpret the data. I would gladly review this again if needed.

Suggestions for improvement

In order of the paper not importance...

- Since there are a lot of experiments on DYRK1A I would like to see that this gene (and maybe some of the other genes) be introduced in the introduction. It's hard to jump into the results without knowing what the genes do. Just a line or two would be sufficient.
- The method of quantifying the ENCC migration into the gut is appropriate. But in addition to the basic migration affect the authors might be missing some patterns that could add depth. If you look at the close ups you can see that there are possible differences in the pattern of aberrant migration. For example, in the *chd2* and *chd8* crispants look very similar with a pattern of ENCCs along the anterior and dorsal aspects of the gut. Is this a real trend among other crispants? I would like to see in the supplement (or as a link to a figure data sharing resource (e.g. figshare.com)) of side by side "family portraits" of additional representative images to show whether patterns exist. It would be interesting if genes with shared functions showed similar patterns which could lead to mechanistic insights. If there is (or isn't) any trends in patterns a line or two to address this in the results section would be helpful for readers and for designing hypotheses for future experiments.
- In figure 2 the authors use crispr/cas9 as the method to knockout/down genes. I think it should be noted in brief in the results the caveats of this method in *Xenopus*. F0 crispants can be mosaic and therefore some cells may not have the gene knocked out and can maybe migrate more easily. We also know that neural crest cells undergo collective migration, and one "normal" cell might carry along the abnormal cell and vice versa. It is great that in the next section the authors show multiple methods that knockdown *Dyrk1a* and not surprisingly the morpholino is most effective since it is specific and more uniformly targets the mRNA, so you would have less cells that can breakaway. The mosaic nature of crisprs in F0 embryos could cloud some of the migration patterns (that may exist) in the previous point.
- In figure 3 A,B the authors zone in on effects of *Dyrk1a*. In the figure the *Dyrk1a* inhibitor shown is TG003. This has not been traditionally used as a *Dyrk1a* antagonist in aquatic models, has it? Maybe some references can be added if so. On the Tocris website it says it primarily targets CLK family members and also DYRK1A and DYRK1B. Harmine on the other hand is listed as a DYRK1A antagonist and is likely more specific. I suggest replacing the TG003 with Harmine in the main figure and showing TG003 in the supplement, or alternatively showing both in the main figure.
- The feeding/excrement experiment is great and really gets into the translational consequences. There is one caveat though. From the experiment how do you know whether or not the decreased excrement is just due to decreased feeding rates? Certainly, decreased *Dyrk1a* function affects craniofacial development which could in turn affect feeding ability. I would like to see a quick experiment where the fluorescent food left in the dish is also counted (a plate reader might work quickly and effectively). This could just be a quick experiment only focused on feeding rates that would go in the supplement.

- The last part connecting serotonin with Dyrk1 is brilliant. However, the connection with the ENCC migration is not really made in these experiments. It would be a huge benefit to show whether the SSR1 could rescue the ENCC migration using the phox2b in situs. This would really tie the story together. As it stands now the SSR1 rescue of Dyrk1a inhibition could be explained by increasing feeding rates, improving craniofacial development or a whole host of other effects that have nothing to do with ENCC development.
- The lab publishing this work has also shown that decreased Dyrk1a in the brain causes abnormal cell cycle and cell death. Can the authors comment on whether the ENCCs are possibly undergoing apoptosis before or as they migrate. This could just be something brief in the discussion maybe around line 236.

Very Minor Suggestions

- One of the most important experiments in this paper are those using inhibitors over different time windows. This gives a peak into when dyrk1a is important for ENCC migration and is one clear advantage to using an aquatic model organism which could be highlighted. I think panels B and C of S3 should be taken out of supplement and added to the main figure 3 since it is a key experiment. It shows that this migration defect is not just a consequence of earlier effects such as gastrulation and also occurs despite the fact that dyrk1a is also required separately for neural crest specification.
- Can you change the red in the figures to another color for people who see color differently. I have trained 2 grad students now who see red and green as shades of grey and would have a hard time with the figures.
- In figure 2A can you label the other staining in brain and peripheral nervous system. I know this is not the focus, but I couldn't help but wonder about this.
- Also, in the figure 3D could the black and white images of fluorescent beads be shown in color so that they could be seen more easily. Or zoom in on representative regions like in the gut in situs. On my laptop and having some sight challenges myself they looked like white boxes with nothing in them.
- In the discussion lines 236-238 gene dosage is mentioned as a future direction. I would argue though that none of the methods used are complete knockouts and thus all the experiments where decreased gene dosage has indeed been probed. The pictures of the whole embryos support this, they are obviously not smaller or have major gut morphology issues that might be expected in strong or complete knockouts. This might be useful to add to the discussion and really makes the work more translational.

Reviewer #2

(Remarks to the Author)

In this study, McCluskey et al. identified the high-risk ASD genes affects enteric neuronal motility using diploid frog *Xenopus tropicalis* model and SSRIs as candidates that ameliorate such defects. These findings not only provide mechanistic insights into gastrointestinal symptoms in ASD patients, but also potential therapeutic strategies to the patients. The study overall is innovative and stringent, and experiments are well-controlled and performed both in vivo and in vitro. The use of *Xenopus* as a model is nice and the focus on intestinal complications is important and understudied. There are minor points that could enhance the robustness of the conclusions.

1. Does CRISPR/Cas9 mutagenesis of hcASD gene and dyrk1a inhibitor treatment affect ENCC viability comparing to control? Figures 2 and 3 show decreased "area of gut migration" in the treatment groups. It would be good to clarify whether increased cell death plays a role in this phenotype. If possible, quantification using a cell death marker staining in ENCC might be relevant.
2. Could SSRI also rescue the defects on gut motility in other genetic background, e.g. SYNGAP1 mutant? The same experiment in Figure 4C and D in an additional CRISPR/Cas9 perturbation background can help address this question. While these would be good to have, they should not preclude publication, especially #2.

Reviewer #3

(Remarks to the Author)

This paper by McCuskey et al. tackles a fundamental question in autism spectrum disorder (ASD) research, namely: what mechanism underlies the frequent gastrointestinal symptoms experienced by this population? McCuskey et al. reveal that many of the high confidence genes associated with ASD are expressed in neural crest-derived neurons of the gut, and that their perturbation causes gut innervation defects in an animal model. The central finding is highly significant, introducing a mechanistic framework for understanding the molecular basis of enteric symptoms in ASD patients, and also introducing SSRIs as a potential therapeutic strategy. The specific investigational strategy was to query the role of five autism spectrum disorder (ASD) risk genes in enteric nervous system development in *Xenopus* tadpoles. First, the authors identified elevated expression of a cohort of 252 high-confidence ASD risk genes in the human enteric nervous system (ENS). Out of this cohort, researchers identified 16 genes that they found to be associated with GI symptoms in individuals with ASD. From these, they focused on 5 that play disparate roles in cell function such as neurotransmission (SYNGAP1 and SCN2A), chromatin regulation (CHD2 and CHD8) and kinase activity (DYRK1A). Each gene was knocked out using CRISPR in *Xenopus tropicalis* tadpoles and found that knockout of any gene limited ENS progenitor migration. They chose to conduct additional experiments with DYRK1A and were able to show that knockout of DYRK1A decreases gastric motility, a phenotype that is ameliorated by SSRI. Although the role for ASD high confidence genes in enteric neurons is not entirely unprecedented, since similar findings were shown for chd8 in zebrafish, I think the potential overall significance of this study is high, because it suggests a generalizable mechanism shared by multiple genes. To that end, I think a couple of the experiments done with DYRK1 should also be done with at least one other gene, so that the generalizability of the findings, and the manuscript title, are better justified. I also think the authors need to include some consideration of other treatments

that have been tried for treating GIS in ASD patients. Otherwise, I think the methodology is sound and sufficiently detailed. My suggestions are noted below.

Major comments and critical experiments:

1. To me, the impact of the paper derives substantially from the insight that many hcASD genes may similarly disrupt enteric neuron migration and gut motility. This is also implied by the title. For that reason, I think it is important to show that loss of at least one other gene from Figure 2C shares core features with DYRK1A loss of function (testing them all is not necessary). Specifically:

1a. Does loss of function of at least one other of those genes change sox10 expression, implicating a shared role in neural crest specification? Either answer in this case is useful.

1b. Does loss of function of at least one other of those genes cause changes in gut motility as assayed in Figure 3C? This is, I think, the most important thing to know.

2. Is there any rescue effect of SSRIs or 5-HT_{2A} agonists on tadpoles with whole-embryo knockout or gene depletion of DYRK1A, and how does the rescue efficacy contrast with rescue of pharmacologically inhibited DYRK1A? This is important to know because patients will presumably experience the pleiotropic effects of their allele in all tissues at all developmental stages. Any answer is fine, it would just be helpful to the interpretation to know.

3. I think it's important that some of the other mechanisms proposed to contribute to gut function in ASD patients, such as developmental delay and a disrupted gut/immune axis, be introduced, and the way they might interface with this ENCC mechanism discussed. I'm particularly wondering about some of the work showing microbiota transfer can rescue GI symptoms in ASD (Kang et al Sci. Reports 2019). Would any interaction between these mechanisms be expected?

Desired experiments:

1. Include scRNA-Seq data for each of the genes in Fig 2C, as done in supplemental fig 2 for dyrk1a.

2. Ideally, if pharmacological inhibitors exist for any of the other genes in Figure 2C, the effect of a pharmacological knockdown versus a whole embryo depletion on ENCC migration can be compared, as done for DYRK1A. But this is not needed if such reagents are not available.

3. Can loss of function of any other of those genes be rescued by an SSRI or 5-HT_{2A} agonist as shown in Figure 4 C/D? This is useful to understanding the scope of potential therapeutic utility of these agents.

Minor Comments:

1. Figure 1C: needs to be cited in text.

2. Figure 1D: If possible, addition of the family statistics such as is included in 1C would serve as a useful comparison. However, I don't think it's necessary to the figure if not available.

3. I believe that the data included in Figure S3C-D is pertinent to the main storyline and should be included in the main figure. If space is limited, I would suggest it be switched out with Fig. 3A-B

4. Figure 4A: Larger text

5. Because each of the CRISPRs and MOs used here have previously been validated for their loss of function efficacy (Willsey et al 2020; 2021), I do not think it is necessary to replicate that work here. However, I do suggest making it more clear in the Results text that these are loss of function reagents that have been previously validated.

Version 1:

Reviewer comments:

Reviewer #1

(Remarks to the Author)

I believe the authors have addressed all of my comments sufficiently in their revisions. I recommend the manuscript to be published.

Reviewer #2

(Remarks to the Author)

I appreciate the authors tried to address my questions. I also understand it is not technically possible for these experiments to be done at this time.

Reviewer #3

(Remarks to the Author)

I think the revised manuscript has made this very interesting paper even better, and it will be valuable to the field through publication in Nature Communications.

The authors have brought considerable effort and rigor to addressing all of my comments as well as those of the other reviewers. I appreciate all the additions they have been able to make.

Because a couple of my suggested experiments weren't possible, I want to confirm specifically that I think those suggestions

were still thoroughly addressed. I had asked whether some of the effects noted with both genetic and pharmacological perturbations of dyrk1a could be recapitulated with similar perturbations of other genes. The authors have tried rigorously to investigate the question, but have been prevented by the severity of pleiotropic defects with perturbations of syngap. I still think knowing whether the conclusions from dyrk are generalizable is important, but the experimental challenges are totally understandable and shouldn't be a hindrance to publication. The scope of the conclusion is framed correctly and doesn't extend beyond their data, and the authors discuss potential strategies for investigating this question as future directions. Looking at the hets in future studies, as they suggest, will be interesting.

Dear Reviewers,

Thank you for your thoughtful comments and suggestions. We feel the manuscript is much improved thanks to your efforts. We appreciate your kind praise and positive outlook on publication. Below please find a point-by-point response addressing your concerns. We appreciate your help improving the manuscript.

Best,

Helen

Reviewer #1 (Remarks to the Author):

This paper by McCluskey and colleagues hits on all the aspects of a great paper. From human gene-phenotype data to mechanistic and behavioral data in an effective animal model. I believe this manuscript should be published in this journal with no hesitation since it is original and of great significance to understanding comorbidities associated with autism. Moreover, this work also contributes to our basic understanding of innervation of the digestive tract. With a few simple experiments and modifications to the writing I believe this paper to be of high impact and a great addition to the autism and developmental biology fields. I have made some suggestions that could elevate, add a bit more depth and help readers interpret the data. I would gladly review this again if needed.

Thank you so much for this kind assessment of our work and for your effort delineating areas for improvement.

Suggestions for improvement

In order of the paper not importance...

- Since there are a lot of experiments on DYRK1A I would like to see that this gene (and maybe some of the other genes) be introduced in the introduction. It's hard to jump into the results without knowing what the genes do. Just a line or two would be sufficient.

Thank you for this suggestion. In the revised manuscript we now discuss known functions for these genes in the introduction (LINES 49-51). We state: "These five genes include several with functional annotations in neurotransmission (*SYNGAP1*, *SCN2A*), gene expression regulation (*CHD8*, *CHD2*) and kinase activity (*DYRK1A*)."

- The method of quantifying the ENCC migration into the gut is appropriate. But in addition to the basic migration affect the authors might be missing some patterns that could add depth. If you look at the close ups you can see that there are possible differences in the pattern of aberrant migration. For example, in the *chd2* and *chd8* crispants look very similar with a pattern of ENCCs along the anterior and dorsal aspects of the gut. Is this a real trend among other crispants? I would like to see in the supplement (or as a link to a figure data sharing resource (e.g. figshare.com)) of side by side "family portraits" of additional representative images to show whether patterns exist. It would be interesting if genes with shared functions showed similar patterns which could lead to mechanistic insights. If there is (or isn't) any trends in patterns a line or two to address this in the results section would be helpful for readers and for designing hypotheses for future experiments.

In the revised manuscript we now show 3 representative images per condition (new Fig. S3). Aside from ENCCs generally localized near the hindbrain area, we do not observe reproducible differences in migration patterns.

- In figure 2 the authors use crispr/cas9 as the method to knockout/down genes. I think it should be noted in brief in the results the caveats of this method in *Xenopus*. F0 crispants can be mosaic and therefore some cells may not have the gene knocked out and can maybe migrate more easily. We also know that neural crest cells undergo collective migration, and one "normal" cell might carry along the abnormal cell and vice versa. It is great that in the next section the authors show multiple methods that knockdown *Dyrk1a* and not surprisingly the morpholino is most effective since it is specific and more uniformly targets the mRNA, so you would have

less cells that can breakaway. The mosaic nature of crisprs in F0 embryos could cloud some of the migration patterns (that may exist) in the previous point.

We now comment on this important point in the results section (LINES 180-185), saying “CRISPR/Cas9 gene editing in *Xenopus* F0 embryos often results in a mosaic, less severe phenotype³⁴, and it is possible that this mosaicism may result in unedited cells mitigating the migration phenotype since enteric neural crest-derived cells have been shown to migrate in chains³⁵.”

- In figure 3 A,B the authors zone in on effects of Dyrk1a. In the figure the Dyrk1a inhibitor shown is TG003. This has not been traditionally used as a Dyrk1a antagonist in aquatic models, has it? Maybe some references can be added if so. On the Tocris website it says it primarily targets CLK family members and also DYRK1A and DYRK1B. Harmine on the other hand is listed as a DYRK1A antagonist and is likely more specific. I suggest replacing the TG003 with Harmine in the main figure and showing TG003 in the supplement, or alternatively showing both in the main figure.

We have previously published TG003 as a DYRK1A inhibitor in *X. tropicalis* in the context of brain development, where we see it phenocopies both Harmine and pro-INDY (Willsey et al., Neuron 2021, Fig. S4). We have added this reference to the main text (LINE 197). The same experiment with harmine is presented in Fig. S5C-D and we observe the same effect as TG003. We chose TG003 as the focus of this study since it lacks the potential off-target of monoamine oxidase that harmine has, since MAO inhibitors potentially also influence gut transit by breaking down serotonin and other neurotransmitters (Fiedorowicz and Swartz, J Psychiatr Pract 2004).

- The feeding/excrement experiment is great and really gets into the translational consequences. There is one caveat though. From the experiment how do you know whether or not the decreased excrement is just due to decreased feeding rates? Certainly, decreased Dyrk1a function affects craniofacial development which could in turn affect feeding ability. I would like to see a quick experiment where the fluorescent food left in the dish is also counted (a plate reader might work quickly and effectively). This could just be a quick experiment only focused on feeding rates that would go in the supplement.

We thank the reviewer for bringing up this point, as it was one we also worried about. Indeed, we show that DYRK1A is required for cranial neural crest development in early embryonic stages (stage 15, Fig. S5A), so we tried to circumvent some of these potential issues by applying the DYRK1A inhibitor later (stage 20 onward). Additionally, we know these drug-treated animals ingest similar amounts of food compared to controls because in the rescue experiments with Lexapro, they excrete similar numbers of beads (Fig. 4C-D), showing they ingested a similar amount compared to the control animals. (The lexapro was added acutely, after feeding, so it could not have affected feeding).

- The last part connecting serotonin with Dyrk1 is brilliant. However, the connection with the ENCC migration is not really made in these experiments. It would be a huge benefit to show whether the SSR1 could rescue the ENCC migration using the phox2b in situs. This would really tie the story together. As it stands now the SSR1 rescue of Dyrk1a inhibition could be explained by increasing feeding rates, improving craniofacial development or a whole host of other effects that have nothing to do with ENCC development.

Thank you for your kind words. We regret that it was unclear that the SSRI addition was acute and after feeding, rather than over development. We have now clarified this in the main text (LINES 242-248): “Acute exposure to either of these compounds only during the excretion portion of the assay (3 hours) rescued the decreased motility phenotype of the Dyrk1a inhibitor-treated animals, restoring excretion to that comparable to the DMSO control (Fig. 4C-D)”.

To be clear, the animals were raised in the DYRK1A inhibitor and then the SSRI was added only during the excretion part of the assay (3 hours). So we do not think the SSRI rescued the phenotype via a developmental mechanism. We chose this experimental strategy as it more closely mimics a clinical setting where the

phenotype would be established and later ideally there would be an acute treatment, since in utero interventions are not currently possible.

- The lab publishing this work has also shown that decreased Dyrk1a in the brain causes abnormal cell cycle and cell death. Can the authors comment on whether the ENCCs are possibly undergoing apoptosis before or as they migrate. This could just be something brief in the discussion maybe around line 236.

We appreciate this point and have added the following sentence to the discussion (LINES 285-287): "In particular, the role of cell death in these phenotypes will need to be explored, as we have previously shown that DYRK1A inhibition in the central nervous system causes apoptosis (Willsey et al 2021)."

Very Minor Suggestions

- One of the most important experiments in this paper are those using inhibitors over different time windows. This gives a peak into when dyrk1a is important for ENCC migration and is one clear advantage to using an aquatic model organism which could be highlighted. I think panels B and C of S3 should be taken out of supplement and added to the main figure 3 since it is a key experiment. It shows that this migration defect is not just a consequence of earlier effects such as gastrulation and also occurs despite the fact that dyrk1a is also required separately for neural crest specification.

Thank you for this comment; we regret that the order in which we presented the data made it confusing. Indeed, the drug experiment presented in Fig. 3 was also done with drug treatment starting after neural crest development (stage 25). We have reordered this section (LINES 194-202) to emphasize that the concern about crest development was with the CRISPR and morpholino experiments, but that the drug treatments can resolve this issue (with Fig. S5B-C showing a schematic of the experiment and other drug controls). For this reason, we have left the figures unchanged.

- Can you change the red in the figures to another color for people who see color differently. I have trained 2 grad students now who see red and green as shades of grey and would have a hard time with the figures.

Thank you for highlighting this important issue. We have now consulted accessibility resources to ensure that our selections are accessible and have changed all figure colors to those within accessible palettes. Thank you for bringing this to our attention.

- In figure 2A can you label the other staining in brain and peripheral nervous system. I know this is not the focus, but I couldn't help but wonder about this.

We have now labeled these in figure 2A and added an explanation in the figure legend. We thank the reviewer for this clarifying addition.

- Also, in the figure 3D could the black and white images of fluorescent beads be shown in color so that they could be seen more easily. Or zoom in on representative regions like in the gut in situs. On my laptop and having some sight challenges myself they looked like white boxes with nothing in them.

We have zoomed in on these images and cropped them to the same region for all images to maintain comparability. We agree that this is easier to see and interpret now.

- In the discussion lines 236-238 gene dosage is mentioned as a future direction. I would argue though that none of the methods used are complete knockouts and thus all the experiments where decreased gene dosage has indeed been probed. The pictures of the whole embryos support this, they are obviously not smaller or have major gut morphology issues that might be expected in strong or complete knockouts. This might be useful to add to the discussion and really makes the work more translational.

We thank the reviewer for pointing this out. We agree that our experiments are loss of function and likely not nulls. Nevertheless, we do think that studying germline heterozygous animals would be of value. We have

edited this sentence accordingly (LINES 287-289): “Additionally, our work here characterizing loss of function for these genes could be complemented by the use of heterozygous germline mutant animals to recapitulate the haploinsufficiency present in individuals with pathogenic *de novo* variants in these genes.”

Reviewer #2 (Remarks to the Author):

In this study, McCluskey et al. identified the high-risk ASD genes affects enteric neuronal motility using diploid frog *Xenopus tropicalis* model and SSRIs as candidates that ameliorate such defects. These findings not only provide mechanistic insights into gastrointestinal symptoms in ASD patients, but also potential therapeutic strategies to the patients. The study overall is innovative and stringent, and experiments are well-controlled and performed both in vivo and in vitro. The use of *Xenopus* as a model is nice and the focus on intestinal complications is important and understudied. There are minor points that could enhance the robustness of the conclusions.

Thank you for your thoughtful comments and suggestions for improving this manuscript.

1. Does CRISPR/Cas9 mutagenesis of *hcASD* gene and *dyrk1a* inhibitor treatment affect ENCC viability comparing to control? Figures 2 and 3 show decreased “area of gut migration” in the treatment groups. It would be good to clarify whether increased cell death plays a role in this phenotype. If possible, quantification using a cell death marker staining in ENCC might be relevant.

We attempted this experiment with the *dyrk1a* morpholino injection, followed by HCR for *phox2b* and cleaved caspase 3 antibody staining. However, we were unable to reliably call whether a dying cell was *phox2b+* or not, and since we did not have confidence in the results, we have not added it here. Instead, we have acknowledged the need to study the role of apoptosis in these phenotypes in the discussion (LINES 287-289): “In particular, the role of cell death in these phenotypes will need to be explored, as we have previously shown that DYRK1A inhibition in the central nervous system causes apoptosis (Willsey et al 2021)”.

2. Could SSRI also rescue the defects on gut motility in other genetic background, e.g. SYNGAP1 mutant? The same experiment in Figure 4C and D in an additional CRISPR/Cas9 perturbation background can help address this question. While these would be good to have, they should not preclude publication, especially #2.

We also attempted this experiment for *syngap1*, but observed severe edema (including swelling of the gut) in the tadpoles. Some animals also had mouth structural defects. Therefore we did not have confidence that the severity of these morphological phenotypes would allow us to make reliable conclusions about feeding and excretion. In the case of *dyrk1a*, the ability to use pharmacological inhibitors allowed us to bypass early structural mouth defects and just focus on the contribution of ENCC migration defects. Therefore, we regret that we are not able to include the requested experiments. We have noted this in the discussion now as an important future direction (LINES 309-310): “It will be important future work to determine the extent to which perturbations of other *hcASD* genes also result in gut dysmotility, and whether serotonin activation can also ameliorate any observed phenotypes.”

Reviewer #3 (Remarks to the Author):

This paper by McCuskey et al. tackles a fundamental question in autism spectrum disorder (ASD) research, namely: what mechanism underlies the frequent gastrointestinal symptoms experienced by this population? McCuskey et al. reveal that many of the high confidence genes associated with ASD are expressed in neural crest-derived neurons of the gut, and that their perturbation causes gut innervation defects in an animal model. The central finding is highly significant, introducing a mechanistic framework for understanding the molecular basis of enteric symptoms in ASD patients, and also introducing SSRIs as a potential therapeutic strategy. The specific investigational strategy was to query the role of five autism spectrum disorder (ASD) risk genes in enteric nervous system development in *Xenopus* tadpoles. First, the authors identified elevated expression of a cohort of 252 high-confidence ASD risk genes in the human enteric nervous system (ENS). Out of this cohort, researchers identified 16 genes that they found to be associated with GI symptoms in individuals with ASD. From these, they focused on 5 that play disparate roles in cell function such as neurotransmission (*SYNGAP1* and *SCN2A*), chromatin regulation (*CHD2* and *CHD8*) and kinase activity (*DYRK1A*). Each gene was knocked out using CRISPR in *Xenopus tropicalis* tadpoles and found that knockout of any gene limited ENS progenitor migration. They chose to conduct additional experiments with *DYRK1A* and were able to show that knockout of *DYRK1A* decreases gastric motility, a phenotype that is ameliorated by SSRI. Although the role for ASD high confidence genes in enteric neurons is not entirely unprecedented, since similar findings were shown for *chd8* in zebrafish, I think the potential overall significance of this study is high, because it suggests a generalizable mechanism shared by multiple genes. To that end, I think a couple of the experiments done with *DYRK1* should also be done with at least one other gene, so that the generalizability of the findings, and the manuscript title, are better justified. I also think the authors need to include some consideration of other treatments that have been tried for treating GIS in ASD patients. Otherwise, I think the methodology is sound and sufficiently detailed. My suggestions are noted below.

Thank you for your positive assessment of this manuscript and for your time and energy improving it.

Major comments and critical experiments:

1. To me, the impact of the paper derives substantially from the insight that many hcASD genes may similarly disrupt enteric neuron migration and gut motility. This is also implied by the title. For that reason, I think it is important to show that loss of at least one other gene from Figure 2C shares core features with *DYRK1A* loss of function (testing them all is not necessary). Specifically:

1a. Does loss of function of at least one other of those genes change *sox10* expression, implicating a shared role in neural crest specification? Either answer in this case is useful.

Thank you for this suggestion. We tested the effects of knocking down the other genes from Figure 2C on *sox10* expression and observe significant defects following depletion of *chd8* or *chd2*. We observe a moderate effect for *syngap1* and no effect for *scn2a*. These results are now included as Fig. S6.

1b. Does loss of function of at least one other of those genes cause changes in gut motility as assayed in Figure 3C? This is, I think, the most important thing to know.

Previous work in zebrafish has shown gut motility defects for *chd8* and *shank3* (referenced in LINES 39, 279-280). Nevertheless, we attempted these experiments for *syngap1*, but with the strong loss of function caused by our genetic tools, we also observed significant edema and craniofacial structural changes (in some cases, no mouth) in the affected tadpoles (consistent with the moderate neural crest defects noted above). Because of this, we did not feel confident assessing gut transit in this context. In the case of *dyrk1a*, the ability to use pharmacological inhibitors allowed us to bypass early structural mouth defects and just focus on the later contributions of the gene to gut function. Therefore, we regret that we are not able to include the requested experiments.

We think an important future direction would be to do this work using germline heterozygous mutant lines, which may have less craniofacial defects compared to our severe loss of function animals. We have noted this in the discussion now as an important future direction (LINES 287-289): “Additionally, our work here characterizing loss of function for these genes could be complemented by the use of heterozygous germline mutant animals to recapitulate the haploinsufficiency present in individuals with pathogenic *de novo* variants in these genes.” We also note (309-310): “It will be important future work to determine the extent to which perturbations of other hcASD genes also result in gut dysmotility, and whether serotonin activation can also ameliorate any observed phenotypes.”

2. Is there any rescue effect of SSRIs or 5-HTR5 agonists on tadpoles with whole-embryo knockout or gene depletion of DYRK1A, and how does the rescue efficacy contrast with rescue of pharmacologically inhibited DYRK1A? This is important to know because patients will presumably experience the pleiotropic effects of their allele in all tissues at all developmental stages. Any answer is fine, it would just be helpful to the interpretation to know.

Again, we attempted these experiments, but because of the severe edema and craniofacial defects following whole-embryo knockout/depletion, we were unable to test this reliably. We agree that this is an important future direction and have commented on it in the discussion, as mentioned in the previous point.

3. I think it's important that some of the other mechanisms proposed to contribute to gut function in ASD patients, such as developmental delay and a disrupted gut/immune axis, be introduced, and the way they might interface with this ENCC mechanism discussed. I'm particularly wondering about some of the work showing microbiota transfer can rescue GI symptoms in ASD (Kang et al Sci. Reports 2019). Would any interaction between these mechanisms be expected?

We agree that considering all mechanisms of GI dysfunction in ASD is important. We have now commented on the need to intersect our findings with other proposed mechanisms including the gut microbiome, particularly since it is known that dysregulated ENS development can alter the gut microbiome (Rolig et al 2017, PMID 28207737). Now we comment in the discussion (LINES 289-291): “Intersecting these findings regarding ENS development with other proposed mechanisms of GI dysmotility in ASD will be important future work, particularly since ENS developmental defects are known to impact the composition of the gut microbiome (Rolig et al., 2017, Kang et al., 2019).”

Desired experiments:

1. Include scRNA-Seq data for each of the genes in Fig2C, as done in supplemental fig 2 for *dyrk1a*.

We have now included the UMAP expression for each of the other genes as Fig S2.

2. Ideally, if pharmacological inhibitors exist for any of the other genes in Figure 2C, the effect of a pharmacological knockdown versus a whole embryo depletion on ENCC migration can be compared, as done for DYRK1A. But this is not needed if such reagents are not available.

We were unable to identify any other specific pharmacological inhibitors to do this.

3. Can loss of function of any other of those genes be rescued by an SSRI or 5-HTR5 agonist as shown in Figure 4 C/D? This is useful to understanding the scope of potential therapeutic utility of these agents.

Again, because these perturbations were severe and caused craniofacial defects, we were unable to do this experiment.

Minor Comments:

1. Figure 1C: needs to be cited in text.

This figure has now been cited in the text (LINE 118)

2. Figure 1D: If possible, addition of the family statistics such as is included in 1C would serve as a useful comparison. However, I don't think it's necessary to the figure if not available.

While 1C is from Simons Searchlight data, which includes family data, Figure 1D is from Ciitizen medical record data, and only includes medical records for individuals with hcASD gene variants (no family). Therefore we cannot include this information because it does not exist.

3. I believe that the data included in Figure S3C-D is pertinent to the main storyline and should be included in the main figure. If space is limited, I would suggest it be switched out with Fig. 3A-B

Thank you for this comment; we regret that the order in which we presented the data made it confusing. Indeed, the drug experiment presented in Fig. 3 was also done with drug treatment starting after neural crest development (stage 25). We have reordered this section (LINES 194-202) to emphasize that the concern about crest development was with the CRISPR and morpholino experiments, but that the drug treatments can resolve this issue (with Fig. S3B-C showing a schematic of the experiment and other drug controls). For this reason, we have left the figures unchanged.

4. Figure 4A: Larger text

We have increased the text size in Figure 4A and agree that it is much more legible now.

5. Because each of the CRISPRs and MOs used here have previously been validated for their loss of function efficacy (Willsey et al 2020; 2021), I do not think it is necessary to replicate that work here. However, I do suggest making it more clear in the Results text that these are loss of function reagents that have been previously validated. Thank you for highlighting this, we have now clarified this in LINES (146-148): "We have previously validated these sgRNAs in *X. tropicalis* for their loss-of-function efficiency (Willsey 2020, 2021)."